# A Potential Therapy Using Antisense Oligonucleotides to Treat Autosomal Recessive Polycystic Kidney Disease

**DOI:** 10.3390/jcm12041428

**Published:** 2023-02-10

**Authors:** Huixia Li, Chunli Wang, Ruochen Che, Bixia Zheng, Wei Zhou, Songming Huang, Zhanjun Jia, Aihua Zhang, Fei Zhao, Guixia Ding

**Affiliations:** 1Department of Nephrology, Children’s Hospital of Nanjing Medical University, Nanjing 210008, China; 2Nanjing Key Laboratory of Pediatrics, Children’s Hospital of Nanjing Medical University, Nanjing 210008, China; 3Jiangsu Key Laboratory of Pediatrics, Nanjing Medical University, Nanjing 210008, China

**Keywords:** splicing variant, *PKHD1*, exon skipping, autosomal recessive polycystic kidney disease, ASOs

## Abstract

(1) Background: Autosomal recessive polycystic kidney disease (ARPKD) is a rare ciliopathy characterized by progressively enlarged kidneys with fusiform dilatation of the collecting ducts. Loss-of-function mutations in the *PKHD1* gene, which encodes fibrocystin/polyductin, cause ARPKD; however, an efficient treatment method and drug for ARPKD have yet to be found. Antisense oligonucleotides (ASOs) are short special oligonucleotides which function to regulate gene expression and alter mRNA splicing. Several ASOs have been approved by the FDA for the treatment of genetic disorders, and many are progressing at present. We designed ASOs to verify whether ASOs mediate the correction of splicing further to treat ARPKD arising from splicing defects and explored them as a potential treatment option. (2) Methods: We screened 38 children with polycystic kidney disease for gene detection using whole-exome sequencing (WES) and targeted next-generation sequencing. Their clinical information was investigated and followed up. The *PKHD1* variants were summarized and analyzed, and association analysis was carried out to analyze the relationship between genotype and phenotype. Various bioinformatics tools were used to predict pathogenicity. Hybrid minigene analysis was performed as part of the functional splicing analysis. Moreover, the de novo protein synthesis inhibitor cycloheximide was selected to verify the degraded pathway of abnormal pre-mRNAs. ASOs were designed to rescue aberrant splicing, and this was verified. (3) Results: Of the 11 patients with *PKHD1* variants, all of them exhibited variable levels of complications of the liver and kidneys. We found that patients with truncating variants and variants in certain regions had a more severe phenotype. Two splicing variants of the *PKHD1* genotypes were studied via the hybrid minigene assay: variants c.2141-3T>C and c.11174+5G>A. These cause aberrant splicing, and their strong pathogenicity was confirmed. We demonstrated that the abnormal pre-mRNAs produced from the variants escaped from the NMD pathway with the use of the de novo protein synthesis inhibitor cycloheximide. Moreover, we found that the splicing defects were rescued by using ASOs, which efficiently induced the exclusion of pseudoexons. (4) Conclusion: Patients with truncating variants and variants in certain regions had a more severe phenotype. ASOs are a potential drug for treating ARPKD patients harboring splicing mutations of the *PKHD1* gene by correcting the splicing defects and increasing the expression of the normal *PKHD1* gene.

## 1. Introduction

Autosomal recessive polycystic kidney disease (ARPKD) is a rare ciliopathy caused by homozygous or compound heterozygous mutations in the polycystic kidney and hepatic disease 1 gene (*PKHD1*). The incidence of ARPKD is estimated to be 1/20,000 of live births. It features progressively growing multiple cysts in the collecting ducts of the kidney and further development to end-stage renal disease (ESRD), but also presents as congenital hepatic fibrosis or even portal hypertension [1,2,3]. ESRD typically occurs in the first 10 years of life and leads to the need for renal replacement therapy before the age of 20 [4] in approximately 50% of cases and progressive hepatobiliary disease requiring a liver transplant in 10% of cases. A variable age of onset has also been reported, ranging from 0.67 to 14 years. The *PKHD1* gene is a large genomic segment that spans ~472 kb, consisting of 4074 amino acids on chromosome 6p12.2, encoding fibrocystin/polyductin (FPC) in the collecting tubules of the kidney. FPC is a ubiquitously expressed single transmembrane (TM)-spanning protein, although it is more abundant in the primary cilium or basal body complex of epithelial cells in the renal tubules and hepatic bile ducts [5,6]. *PKHD1* mutations lead to the extension and dilation of the collecting ducts to form microcysts, further resulting in the progressive enlargement of the kidneys. To date, more than 800 *PKHD1* variants have been reported (Human Gene Variant Database, HGMD, April 2022, date last accessed). The existing disease management mainly consists of symptomatic treatment, the use of RRT therapy and even transplantation at the late stage of the disease.

Gene therapy is increasingly being applied to a variety of diseases [7,8]. Missense or nonsense variants in the *PKHD1* gene are the most common (69.7%). However, the large genomic size of *PKHD1* and the highly intricate proteolytic processing of the FPC protein in the mechanism of the protein translation processes made the study of *PKHD1*/FPC particularly difficult. Among the *PKHD1* gene variants, 7.7% are splicing variants, reminding us of the necessity of studying the pathogenic characteristics of and potential treatments for splicing variants in *PKHD1*. Nucleic acid medicines such as small interfering RNAs (siRNAs) and antisense oligonucleotides (ASOs) have been verified as promising potent drugs for the treatment of incurable genetic disorders and are emerging as some of the most pragmatic therapeutic modalities for the multimodal treatment of genetic and rare diseases [9,10]. Since the discovery of ASOs in 1978, research on small nucleic acid drugs has shown continuous progress [11]. ASOs are single-stranded oligonucleotides which specifically bind to the target mRNA and cause steric hindrance by cleaving the mRNA, by degrading the target’s transcript or by closing the key regions through complementary anti-parallel-based pairing [12,13], which in turn lead to altered mRNA processing and modulated gene expression. Clinical experiments of ASOs have been carried out since 1993, and ASOs have gradually become available and applied to clinical practice since 2016.

In recent years, ASO-based therapies for specific splicing variants in patients have been successfully developed [14]. In 2016, the FDA approved nusinersen, the first ever targeted intrathecal injection therapy using an antisense drug, which is used for spinal muscular atrophy (SMA). It stimulates the production of the functional SMN protein of the spinal column and CNS (including motor neurons) by inducing the retention of Exon 7 of the SMN2 gene, resulting in improvements in motor activity and survival [15,16,17]. ASOs are also useful for generating a partially functional truncated protein by causing exon skipping to restore in-frame reading of the dystrophin gene to achieve a treatment for Duchenne muscular dystrophy (DMD). Eteplirsen, which was approved by the FDA in September 2016, is designed to promote the in-frame skipping of dystrophin’s Exon 51 [18], whereas golodirsen (which was approved in December 2019) is designed to cause the skipping of Exon 53 [19,20], and casimersen causes the skipping of Exon 45 [21].

To date, more than 50 antisense drugs are now in clinical trials worldwide. ASOs’ huge potential is linked to their illness fighting ability. In this research, we attempted to ascertain novel exon-extending targets in *PKHD1* by reporting the relationship between exons’ overexpressed patterns and the rescue of extended mRNA expression in cells. We showed that all ASOs targeting *PKHD1′*s Exon 21, to a great extent, restore the expression of the normal transcript in cells, indicating that these ASOs might be promising therapeutic agents in the treatment of ARPKD caused by Exon 21 splicing mutations of the *PKHD1* gene.

## 2. Materials and Methods

### 2.1. Clinical and Laboratory Data

Our center carried out genetic sequencing screening in 38 children with PKD. We enrolled these children with cysts in their kidneys, and the clinical data, ultrasound findings and laboratory test results were collected in detail. The clinical data included age at diagnosis, ethnicity, phenotypic appearance (renal cysts, kidney enlargement, Caroli disease, hepatic cysts, splenomegaly, etc.), family history and relevant laboratory data. The patients’ parents or legal guardians gave informed consent. The institutional ethics committee of the Children’s Hospital of Nanjing Medical University approved our study before whole-exome sequencing was carried out.

### 2.2. Genetic Analysis

Genetic tests were performed as previously described. Genomic DNA was extracted using a DNA isolation kit (Tiangen, Beijing, China) following the manufacturer’s protocol. We amplified the sequences including the exonic and intron–exon boundaries of *PKHD1* using a polymerase chain reaction (PCR) and designed primers using Primer Premier 5.0 (Palo Alto, CA, USA). The products of PCR were purified using the QIAquick PCR Purification Kit (Qiagen, Shanghai, China) and were then sequenced using the Big Dye Terminator (Applied Biosystems, Foster City, CA, USA) via an ABI 3130 genetic analyzer (Applied Biosystems). All variants were denoted on the basis of the NCBI reference sequence for *PKHD1* (NM_138694.4). Genomic DNA was used for whole-exome sequencing (WES) and a targeted next-generation sequencing test (including 140 short related genes, which are shown in Appendix A). The test was performed on an Illumina HiSeq 2500 sequencing platform for 150 bp paired reads. The sequences were compared with the reference human genome (hg19 build), and the deletions (INDEL), inserts and single nucleotide variations (SNV) were filtered. The variants were annotated and those with allele frequencies of more than 1% were removed by the databases (dbSNP, an in-house MAF database; 1000 Genomes MAF (Chinese); Genome Aggregation Database (gnomAD) and ExAC). All candidate variants were selected via the criteria of the ACMG (American College of Medical Genetics and Genomics) and further verified via Sanger sequencing.

### 2.3. In Silico Analysis

The potential pathogenicity of novel missense variants of *PKHD1* were predicted using bioinformatics tools such as SIFT, PolyPhen-2, Provean, ACMG, VariantTaster and Consurf conservative prediction. Splice Site software was used to predict the latent impact of the consensus 5′ and 3’ splice sites and to predict the generation and/or activation of new splicing sites. We used the MaxEntScan (http://hollywood.mit.edu/burgelab/maxent/Xmaxentscan_scoreseq.html, accessed on 29 March 2021), SpliceAI (https://spliceailookup.broadinstitute.org/, accessed on 29 March 2021) and Human Splicing Finder (http://www.umd.be/HSF3/HSF.shtml, accessed on 29 March 2021). The analysis of the variants with these bioinformatics tools was carried out simultaneously.

### 2.4. Construction of the Plasmid

The creation of hybrid minigene constructs was based on the pSPL3 vector, which contains a conventional expression system with two exons (SD6 and SA2) to further prompt the synthetic mRNA transcript analysis [22]. The sequence of upstream primer SD6 is (5′-3′) TCTGAGTCACCTGGACAACC and the downstream SA2 is (5′-3′) ATCTCAGTGGTATTTGTGAGC. The fragment containing *PKHD1* target exon (22, 47 and 61) fragments where the variants were located and at least 200 bp of their intronic flanking regions with XhoI and BamHI restriction sites were amplified via PCR using specific oligonucleotide primers and the extracted DNA. The web-based source primer (https://www.ncbi.nlm.nih.gov/tools/primer-blast/index.cgi, accessed on 1 April 2021) was used to design the primers, which are shown in the Appendix A. PCR products were purified and sequenced, and then were cloned into the pSPL3 vector via restriction enzymes XhoI and BamHI and the wild-type and mutant minigene recombinants constructed. A hybrid minigene mainly produces two transcripts; one is a larger transcript located in the upper part of the figure (SD6, the inserted exon and SA2), another is a smaller-sized (263 bp) transcript located below (SD6 and SA2) (Figure 1). The obtained wild-type (WT) and mutant plasmids were verified using bi-directional sequencing.

### 2.5. In Vitro Minigene Assay

HEK293T, HeLa, A549 and HepG2 cells were seeded with 10% PAN serum (DMEM/10% FAN) in an environment of 5% CO_2_ at 37 °C until they were 75–80% confluent. The minigenes (wild-type and mutants) of *PKHD1* were transiently transfected into the HEK293T, HeLa, A549 and HepG2 cell lines using Lipofectamine 2000 (Invitrogen, Carlsbad, CA, USA) or PolyJetTM DNA Transfection Reagent (Signagen, Shanghai, China), following the instruction manuals.

Total RNA was extracted after 48 h of transfection and was reverse-transcribed into cDNA. PCR amplification was performed by the primers SD6 and SA2. The size and transcripts of the products of PCR were analyzed after agarose gel electrophoresis and sequenced. The following formula was used to quantify the percentage of aberrant splicing: exclusion percentage (%) = (lower band/(lower band + upper band)) × 100. The error bar represents the SEM (*n* = 3, * *p* < 0.05, unpaired Student’s *t*-test).

### 2.6. Design of Antisense Oligonucleotides

Mfold software was used to analyze the structure of the RNA near the *PKHD1* gene variant c.2141-3T>C and to confirm the open and closed regions. Subsequently, the appropriate sequences of ASOs were chosen, which included nucleotide lengths between 16 and 25, a GC content from 40 to 60% and a Tm higher than 48 °C. In addition, we also verified that the free energy value of the ASO was above −4, the free energy value of the dimer was above −14 and the difference in the energy values between the candidate region and the ASOs’ binding region was 21–28. The precise sequences were UUAAGAUGGUAGACUUGCUGUGUGG (ASO1), AGACUUGCUGUGUGGAAAAUCC (ASO2) and GUGACUUAAGAUGGUAGACUUGCU (ASO3) (Figure 2D). All ASOs with phosphorothioate chains and 2-methyl sugar modifications were synthesized by Tsingke (Tsingke Biological Technology, Beijing, China).

### 2.7. ASOs Rescue Assays

HEK293T cells were cultured and transfected with minigene plasmids. The ASOs were transfected at a concentration of 0.5 μM, 24 h after transfection. Total RNA was extracted after 48 h of transfection for transcriptional analysis via RT–PCR.

## 3. Results

### 3.1. Clinical Analysis

We identified 11 patients who carried homozygous or compound heterozygous *PKHD1* gene variants out of the 38 individuals with kidney cysts. The age of onset of 11 patients (six females and five males) ranged from 0.67 to 14 years (median = 5 years). All patients were identified from the Pediatric Nephrology Department and had been assessed by an experienced sonographer, except for Patient 8 (who was born with cysts). Patients 2, 9 and 11 had a family history of renal cysts (Appendix A). Patients 1 and 6 also presented with inguinal hernia disease. Patients 2 and 9 suffered from multiple recurrent urinary tract infections, and Patient 9 presented with the renal complication of kidney stones. Patient 8 was diagnosed at birth because of their apparent symptoms and signs of peripheral facial paralysis, hypothyroid and atrial septal defects. Moreover, Patient 3, who had the severe hepatic complication of Caroli disease, manifested with hepatosplenomegaly, intrahepatic bile duct dilatation, liver fibrosis, severe portal systemic shunting and portal hypertension, which resulted in an emergency laparoscopic splenectomy in 2020; the patient was accepted for allogeneic kidney transplantation last year. Patient 10 was on a regimen of oral ACEI combined with diuretics, and had long-lasting hematuria, hypertension and mild proteinuria, but normal renal function. The eGFR for all patients was greater than 100 mL·min^–1^·m^–2^, although some had elevated serum creatinine (such as Patient 9 in this study). End-stage renal disease has developed in no patient so far. The clinical features of the 11 patients are shown in Table 1.

### 3.2. Analysis of PKHD1 Variants

We identified 17 *PKHD1* variants (3 nonsense, 10 missense, 1 frameshift and 3 splice sites) in the 11 different pedigrees (Table 2, Appendix A). The missense variants c.325G>A(p.(Ala109Thr)), c.3500T>C(p.(Leu1167Pro)), c.5869G>A(p.(Asp1957Asn)), c.6245C>T(p.(Thr2082Ile)), c.10072G>A(P.(Asp3358Asn)) and c.11525G>A(p.(Arg3842Gln)) had not been reported in the genomic databases or the previous literature. The six novel missense variants p.Ala109Thr, p.Leu1167Pro, p.Asp1957Asn, p.Thr2082Ile, p.Asp3358Asn and p.Arg3842Gln were all predicted to be damaging and deleterious, and probably affected protein function, according to the in silico analysis (Appendix A). At the same time, the variants p.Leu1167Pro, p.Asp1957Asn, p.Asp3358Asn and p.Arg3842Gln were predicted to be highly conservative.

A novel splicing variant, c.11174+5G>A, which is located at +5 bp near the splice donor site in Intron 61 (Figure 1E and Figure 3A), was detected in Patient 3 and his mother. Another novel variant, c.2141-3T>C, was a T/C transition located at −3 bp near the splice acceptor site in Intron 21 in Patient 1 (Figure 1D). The c.11174+5G>A variant should be classified as VUS (PM2 + PP3), and c.2141-3T>C should be classified as VUS (PM2 + PM5 + PP3) according to the ACMG’s standard. The SpliceAI analysis indicated that the variant c.2141-3T>C activated the cryptic splice acceptor site of the intron, whereas c.11174+5G>A may disturb the authentic splice donor site and activate an intronic potential donor site of Intron 61 (Appendix A). Furthermore, four bioinformatic splicing tools (Human Splicing Finder, MaxEntScan, NNSplice and BDGP) predicted that c.2141-3T>C would activate the splice acceptor site and the variant c.11174+5G>A would disrupt the splice donor site (Appendix A). Nephronophthisis (NPHP)-related genes or protein kinase D1 (PKD1) or PKD2 with pathogenic mutations were not detected.

### 3.3. Splicing Analysis of the PKHD1 Minigene

Given the pathogenic potential of the nonclassical splicing variants and the fact that samples from patients are not always available, the functional tests of hybrid minigenes in splicing reporter plasmids, such as pSPL3, have become valuable tools to check the splicing profiles induced by a sequence variation [29,30]. We first cloned the classical donor site splice variant c.7351-2A>T from Patient 1 into the pSPL3 minigene vector. The RT-PCR results showed that the c.7351-2A>T variant induced aberrant splicing patterns (Exon 47 skipping) which confirmed that the pSPL3 minigene reporter was a dependable tool for functionally assaying the potential splice site variants (see Appendix A).

To further identify the pathogenicity of the c.2141-3T>C and c.11174+5G>A variants, wild-type (E22-WT and E61-WT) and mutant (c.2141-3T>C and c.11174+5G>A) minigene vectors were created and transfected to verify the splicing via agarose gel electrophoresis (Figure 1). In the pSPL3 context, the E22-WT construct produced two transcripts; one was a full-length transcript (SD6, E22 and SA2) (402 bp) and the other was a smaller (263 bp) transcript (SD6 and SA2). However, the c.2141-3T>C construct produced a larger transcript of 453 bp including SD6, E22, SA2 and the inserted 51 bp sequences. The E61-WT construct produced a transcript with the expected size (SD6, E61 and SA2) (1281 bp). However, the c.11174+5G>A construct produced a transcript of 263 bp (SD6 and SA2). Subsequent sequence analysis showed that the c.2141-3T>C mutation minigene led to aberrant splicing, as the 51 bp sequence downstream from Intron 21 was restricted (Figure 1). Sequence analysis of the c.11174+5G>A mutation minigene indicated that it lacked the entire Exon 61. HeLa, A549 and HepG2 cells were used to clarify whether this transcriptional change and proportional change in exon exclusion also happened in different cells. Likewise, the c.2141-3T>C mutant minigenes produced a 453 bp transcript in the three cell lines, similar to that in the HEK293 cells after sequencing (see Appendix A). The c.11174+5G>A mutant minigenes produced a transcript that skipped Exon 61.

### 3.4. Treatment with CHX

The variants c.7351-2A>T and c.11174+5G>A were confirmed to have caused the abnormal transcripts, according to the results. The c.2141-3T>C mutant transcript had a 51 bp insert, adding more amino acids to the original length of 714 amino acids and introducing a premature termination codon (PTC) in 719 amino acids (p.V714Afs*5), according to computational prediction analysis (Figure 1). The c.11174+5G>A mutant transcript introduced a frameshift involving a change in the open reading frame with a PTC (p.G3386Efs*10) (Figure 1). The aberrant transcripts with premature termination codons (PTCs) were generated because the splicing and nonsense variants resulted in exon skipping, causing frameshifts in the coding sequence. The abnormal pre-mRNAs with PTCs may be rapidly degraded by the NMD pathway; this has made progress in the fight against many diseases [31,32]. According to the NMD prediction (https://nmdpredictions.shinyapps.io/shiny/, accessed on 17 September 2021) tool, no predictions were available for the non-frameshift variant of c.2141-3T>C; the c.7351-2A>T (p.G2451Vfs*4) and c.11174+5G>A (p.G3386Efs*10) variants were predicted to be subjected to degradation by the NMD pathway (Figure 2A). In fact, c.2141-3T>C was confirmed to have caused the 51 bp insertion, which was likely to lead to the production of PTC, reminding us of the importance of the next experiment.

To clarify whether the transcripts from the mutant minigenes were vulnerable to NMD, we used a de novo protein synthesis inhibitor cycloheximide (CHX), which has been shown to inhibit NMD [33]. Interestingly, after treatment with CHX, the three variants had no effect on the volume or size of these artificial transcripts, suggesting that the c.2141-3T>C, c.7351-2A>T and c.11174+5G>A splicing variants may escape from the control of NMD (Figure 2B,C).

### 3.5. Treatment with ASOs

A classical antisense therapeutic approach to mediate the skipping of the variant exons or the control of mRNA expression is to use ASOs, which are short fragments of single-stranded DNA sequences that target specific exons by recognizing the target’s sequence and binding the pre-mRNA and mRNA to directly alter pre-mRNA splicing, mRNA degradation and gene expression [34,35]. ASOs have been confirmed to specifically increase the expression of productive transcripts and reduce seizures and the incidence of SUDEP as a gene-specific treatment for Dravet syndrome and diastolic function in a murine heart failure model through the inhibition of RBM20 with ASOs in a recent study [36,37]. In vitro minigene analysis verified that exon inclusion had occurred, which may have caused the production of PTC via a frameshift. We supposed that the ASOs may block the pseudoexon insertion associated with variant c.2141-3T>C by modulating the splicing enhancer and modifying the pre-mRNA splicing pattern.

To verify whether ASOs blocked the pseudoexon insertion associated with c.2141-3T>C, two ASOs were designed to block the exonic splicing enhancer and exclude the pseudoexons. We then analyzed productive and nonproductive PKHD1 transcripts and evaluated the influence of ASOs on the level of RNA. We identified that these ASOs significantly increased the expression of their respective unspliced full-length PKHD1 transcripts in HEK293 cells, with a concomitant reduction in the E22-skipping transcript of the WT and c.2141-3T>C in ASO1 and ASO2, while the E22-skipping transcript was unaffected by ASO3 (see Appendix A). Moreover, these ASOs almost completely restored the correct splicing, to a certain extent (Figure 2E).

## 4. Discussion

Herein, we identified 11 patients carrying rare compound heterozygous variants of the *PKHD1* gene. The 17 variants included 3 nonsense variants, 10 missense variants, 1 frameshift and 3 splice sites. The clinical features of our 11 patients were in accordance with the pathophysiology of ARPKD, including multiple cysts, hematuria, hypertension, proteinuria, urinary tract infections, kidney stones, inguinal hernia, hepatic fibrosis, portal hypertension, end-stage renal disease (ESRD) and hepatobiliary disease, etc. Using genetic analysis, all patients were diagnosed with ARPKD.

The clinical phenotypes of ARPKD are highly variable, ranging from severe early-onset antenatal multiple cysts to late-onset hepatic fibrosis, portal hypertension and ESRD [38], similar to the results of a previous report. In this study, Patient 3 had the most severe symptoms, and was admitted for abdominal pain [39], was diagnosed with Caroli’s syndrome and underwent hepatic transplantation because of severe portal systemic shunting and portal hypertension after the subsequent 3-year follow-up period.

The severity of protein dysfunction has been proposed to explain this phenomenon. The *PKHD1* gene encodes fibrocystin, a single-pass membrane protein, which is mainly expressed in the primary cilium of the renal collecting tubules and hepatic bile ducts [40]. Pathogenic mutations in the *PKHD1* gene underlie FPC dysfunction, resulting in a reduced capacity to bind PC2, which, in turn, causes the collecting ducts to elongate and dilate, the formation of microcysts and the diffuse enlargement of the kidney [41,42]. As a consequence, the affected individuals progress to ESRD and hepatobiliary disease in childhood.

We screened six novel missense variants, of which p.Ala109Thr and p.Leu1167Pro were located in part of six highly conserved immunoglobulin-like plexin transcription factor (IPT) domains. These variants mapped to the domains that may impair the function of FPC through their effects on integrins, transcription factors, the regulation of cell separation, motility and the invasion of the extracellular matrices [43,44]. The variants p.Asp1957Asn and p.Thr2082Ile are in the G8 domain and may affect extracellular ligand binding and catalytic activity. The missense variants located in the G8 domains of the human *PKHD1* protein caused an unstable protein because of the incorrect folding of G8 domains, as has been reported by [45]. The variant p.Arg3842Gln is upstream of the ciliary targeting sequence (CST) which allows efficient *PKHD1* trafficking to the cilium by controlling trafficking and affecting glycosylation [46,47]. In silico analysis predicted that these missense variants may affect the function of FPC.

The novel splicing variant c.11174+5G>A was predicted, causing the skipping of Exon 61 by altering the WT donor site, and may introduce a PTC 10 codons downstream, according to the in silico analysis. Our minigene splicing assays verified that c.11174+5G>A resulted in aberrant splicing. The aberrant mRNA transcripts were not degraded by the NMD pathway, as revealed by the administration of the specific de novo protein synthesis inhibitor CHX, indicating that a truncated protein without the C-terminal region including the CST and the nuclear localization signal (NLS) was produced. This reminds us that the variant c.11174+5G>A of ARPKD is likely to be highly pathogenic. Therefore, we upgraded the variant c.11174+5G>A to “likely pathogenic” (PM2 + PP3 + PS3).

The other novel splice site variant c.2141-3T>C may cause the skipping of Exon 22 through obtaining a new WT acceptor site, according to the in silico analysis. Surprisingly, the variant c.2141-3T>C was verified to have inserted an additional 51 bp. We considered that the c.2141-3T>C variant inserted the sequence “AG” to the 5′ original splice acceptor site, which may have deactivated the physiological GT-AG splice sites determining *PKHD1′*s Exon 22 and concurrently activated the cryptic splice sites located in the chimeric Intron 21, ultimately producing an alternative splice acceptor site that was superior to the original one.

The physiological GT-AG splicing site that determines *PKHD1′*s Exon 22 can be inactivated, while the crypto-splicing site within the chimeric Intron 21 can be activated to form an alternative splicing receptor site that is superior to the original receptor site during transcription.

Moreover, aberrant mRNA transcripts may be degraded through the NMD pathway, according to the NMD predictions. However, the small transcriptional products are stable, and the use of de novo protein synthesis inhibitors revealed that c.2141-3T>C was also likely to be highly pathogenic. Thus, we upgraded the variant c.2141-3T>C (p.V714Afs*5) to “likely pathogenic” (PM2 + PM5 + PP3 + PS3).

To date, an efficient disease-modifying therapy for ARPKD has not been approved. Tesevatinib (TSV), as a possible therapy for ARPKD, has yielded promising results in preclinical studies, has obtained approval and is undergoing clinical Phase I and II trials [48]. Renal transplantation is still the last resort for ARPKD with severe ESRD, although the shortage of organs is the most pressing concern. Renal gene therapy is already a suitable alternative to orthotopic renal transplantation for the treatment of inherited renal diseases, such as ASOs [49,50,51]. Splicing mutations cause exon skipping in the *PKHD1* gene encoding the FPC protein, either causing or increasing the severity of ARPKD. Heavy intracellular accumulation of truncated FPC proteins may underlie the more severe symptoms of ARPKD. However, functional studies of FPC and the pathogenic mechanism of diseases have yet to be carried out, potentially limiting the therapeutic effect and viability of these approaches. In light of these findings, reducing the production of exon-free mRNA may provide a disease-modifying therapy for ARPKD. Antisense technology has advanced, and ASOs benefiting patients with various common and rare diseases are increasing [52]. RGLS4326, as part of the microRNA-17 (miR-17) family, was designed to directly bind or inhibit specific miRNAs, displacing miR-17 from translationally active polysomes, and to de-represses multiple miR-17 mRNA targets; this has led to clinical trials for the treatment of ADPKD [53]. Here, ASOs were designed to destroy the novel splicing site that causes inactivation, with no effect on the binding between the auxiliary splicing proteins and splicing enhancer motifs which are related to the function of original splicing. Our results suggested that ASOs may be a new option for ARPKD patients with aberrant splicing, as all of the ASOs could effectively rescue the splicing abnormalities caused by the c.2141-3T>C mutation of the *PKHD1* gene in vitro. Surprisingly, we discovered that ASOs correct the production of abnormal mRNA in HEK293 cells overexpressing *PKHD1*. ASO therapy can potentially overcome some of the limitations of the current approaches by correcting the truncated FPC production by targeting *PKHD1* RNA intracellularly. It is important to further determine the efficacy of ASOs in ARPKD in iPSC-derived ARPKD patient kidneys and ARPKD patients’ tissues, as well as ARPKD animal models.

## 5. Conclusions

Our study found eight novel variants, expanding the genotype of *PKHD1*, and verified that the c.2141-3T>C variant causes a 51 bp insertion upstream from Exon 22, and the c.11174+5G>A variant leads to the skipping of Exon 61 in the mRNA transcript of the *PKHD1* gene. Aberrant transcripts were not degraded through the NMD pathway. This phenomenon means that a potentially pathogenic splicing variant is relevant to renal cysts and congenital hepatic fibrosis. Moreover, the aberrant transcript caused by the c.2141-3T>C variant was rescued by ASOs in vitro, introducing the ASOs as a promising strategy for treating ARPKD patients with intronic variants for the first time.

## Figures and Tables

**Figure 1 jcm-12-01428-f001:**
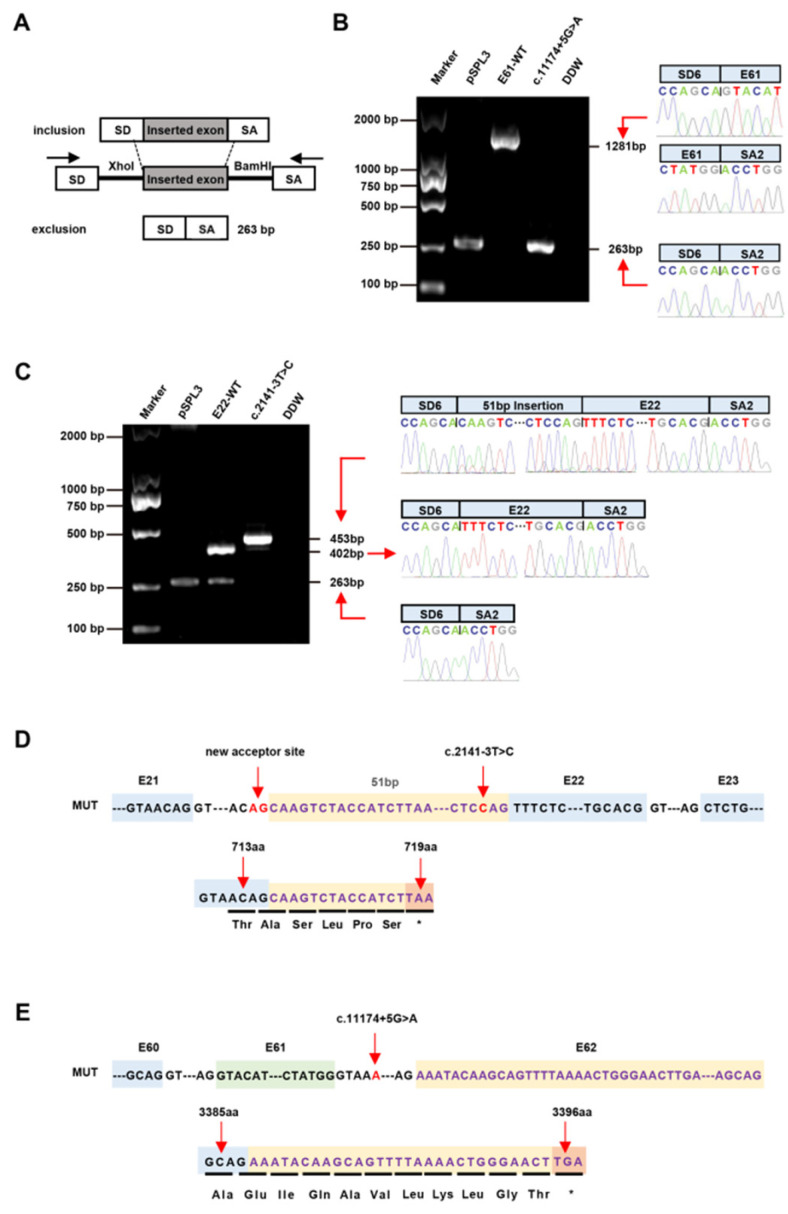
The hybrid minigene assay showed the aberrant mRNA splicing of two splicing variants of the PKHD1 gene in vitro in ARPKD. (**A**) The products of hybrid minigene transcripts in HEK293 cells were amplified by RT-PCR. The transcripts produced by the hybrid minigene are schematically shown, and the arrows show the primers used for amplification (inset) (Wang et al., 2018b). (**B**) Agarose gel electrophoresis and directly sequences (right panel) of the RT–PCR products of the minigene transcripts in HEK293T cells of the variant c.11174+5G>A. cDNA sequencing of variant c.11174+5G>A showed the skipping of Exon 61. Lane 1: marker; Lane 2: pSPL3 (263 bp); Lane 3: E61-WT (1281 bp); Lane 4: c.11174+5G>A (1281 bp and 263 bp); Lane 5: DDW. (**C**) Agarose gel electrophoresis and direct sequences (right panel) of the RT–PCR products of the minigene transcripts in HEK293T cells of the variant c.2141-3T>C. Sequencing of the variant c.2141-3T>C cDNA showed an additional insert of 51 bp between Exon 21 and Exon 22. Lane 1: marker; Lane 2: pSPL3 (263 bp); Lane 3: E22-WT (402 bp and 263bp); Lane 4: c.2141-3T>C (453 bp); Lane 5: DDW. (**D**) Schematic representation of the abnormal splicing of the *PKHD1* exons of variant c.2141-3T>C (numbered boxes). A pseudoexon with a 51 bp insert in the junction of Exon 21 and Exon 22 appears in the mutant transcript (upper panel). The predicted amino acid structures were translated from the aberrant transcripts of the c.2141-3T>C variant with a fragment of 51 bp within Intron 21 in *PKHD1* (lower panel). (**E**) Schematic representation of the abnormal splicing of the *PKHD1* exons of the variant c.11174+5G>A (numbered boxes). Exon 61 was skipped in the mutant transcript (upper panel). The predicted amino acid structures were translated from the aberrant transcripts of the c.11174+5G>A variant with a 1018 bp deletion of Exon 61 in *PKHD1* (lower panel).

**Figure 2 jcm-12-01428-f002:**
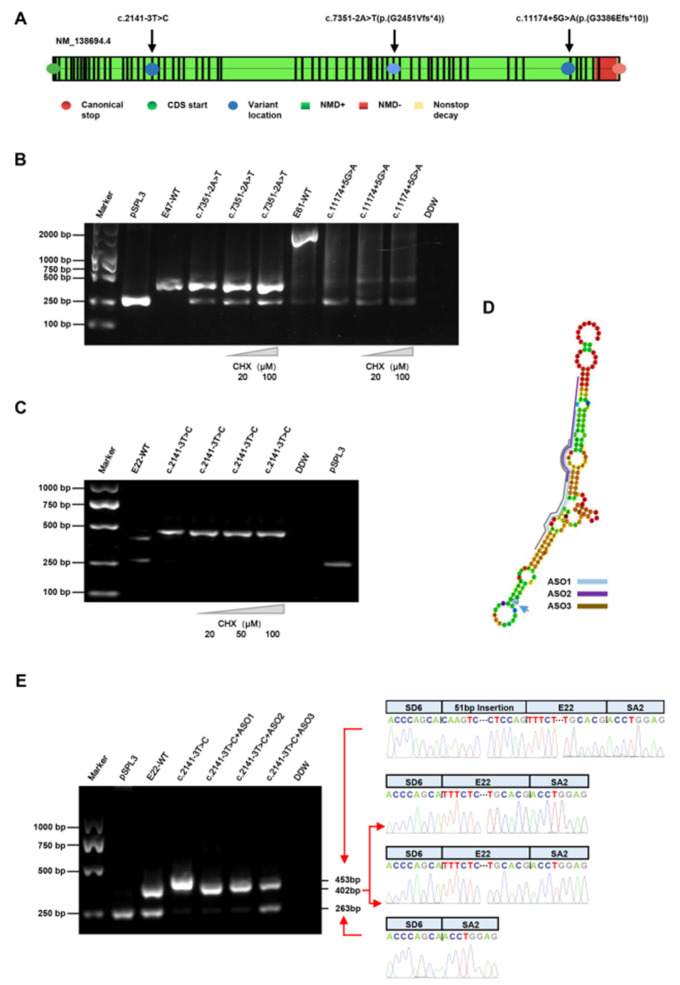
The effect of the de novo protein synthesis inhibitor (cycloheximide) and antisense oligonucleotides (ASOs) on the c.2141-3T>C variant of the PKHD1 gene of aberrant splicing. (**A**) Schematic representation of the *PKHD1* gene showing the localization of the truncating variants subjected to (NMD+) and escaping(NMD-) to nonsense-mediated RNA decay regions (https://nmdpredictions.shinyapps.io/shiny/). The loss-of-function variants p.G2451Vfs*4 and p.G3386Efs*10 detected in this study are shown in red. The region of *PKHD1* where the truncating variants trigger NMD is indicated in green. The region that escaped NMD is represented in red. The nonstop decay region is indicated in yellow. (**B**,**C**) Treatment with cycloheximide (CHX) showed that the transcriptional products were stable. CHX did not degrade the abnormal pre-mRNAs of the splicing variants of the *PKHD1* gene in HEK293T cells. The gray triangle represents the dose-dependent concentration of the reagents. CHX, cycloheximide. (**D**) The positions of three ASOs for the c.2141-3T>C variant in this study. (**E**) RT–PCR analysis of the splicing correction after the ASOs treatments. HEK293T cells were transfected with the wild-type (WT) minigene and the corresponding mutant (MU) minigene containing the variant c.2141-3T>C. With the exception of the nontreated lanes (NT), the remaining lanes show that the administration of three ASOs (ASO1, ASO2, and ASO3) efficiently rescued the retention of the intron caused by c.2141-3T>C. The ASOs rescued the c.2141-3T>C variant of the *PKHD1* gene, which causes abnormal splicing in HEK293T cells.

**Figure 3 jcm-12-01428-f003:**
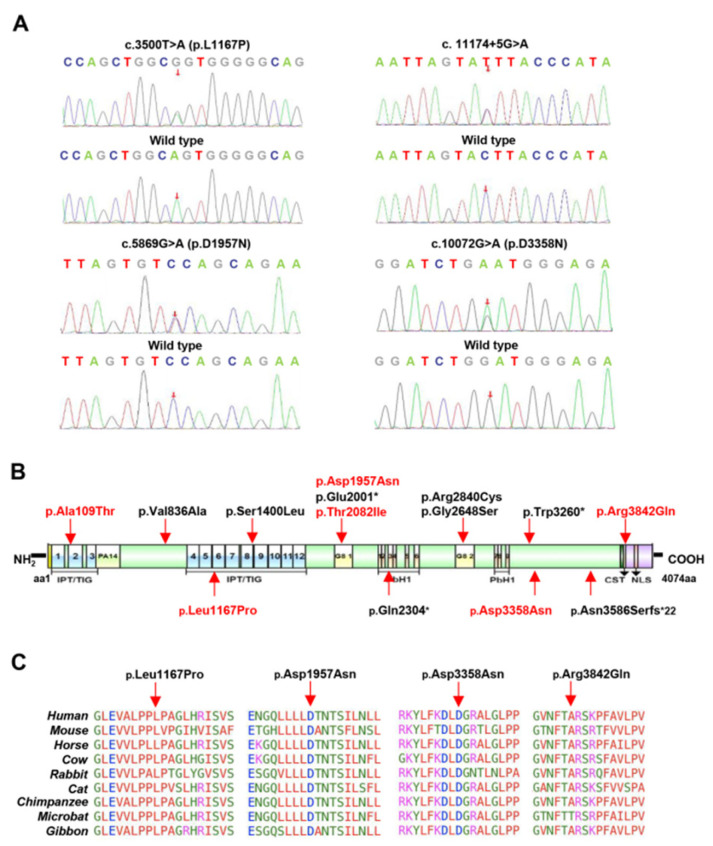
Identification of the PKHD1 variants in patients with cysts. (**A**) Sequence chromatograms of the *PKHD1* variants as detected. (**B**) Distribution of the *PKHD1* variants in corresponding domains. (**C**) The *PKHD1* variants are located in the corresponding domains. Abbreviations: TM domain, transmembrane span; IPT/TIG domain, Ig-like, plexins and transcriptions factors; PA14 domain, Pfam Entry 14; G8 domain, eight conserved glycine residues; PbH1 domain, parallel beta-helix repeats; CST domain, ciliary targeting sequence; NLS domain, nuclear localization signal. The red arrow is points to the orthologous and paralogous protein alignments showing the high conservation of each amino acid altered by four missense variants from humans to gibbon.

**Table 1 jcm-12-01428-t001:** Clinical features of 11 *PKD* patients with the *PKHD1* variants.

Patient	Age (Year)	Sex	Ethnicity	Family History	Renal Cysts	Kidney Enlargement	Caroli Disease	Hepatic Cysts	Splenomegaly	Renal Dysfunction	Liver Dysfunction
P1	7	Female	Han	No	Yes	Yes	No	No	No	No	No
P2	1.6	Female	Han	No	Yes	No	No	No	No	No	No
P3	9	Male	Han	Yes	Yes	Yes	Yes	Yes	Yes	No	Elevated liver enzymes
P4	14.3	Female	Han	No	Yes	No	No	No	No	No	No
P5	0.7	Male	Han	No	Yes	No	No	No	No	No	No
P6	1.6	Male	Han	No	Yes	No	No	No	No	No, with urinary occult blood	No
P7	1	Male	Han	No	Yes	Yes	No	No	No	No	No
P8	0.8	Female	Han	No	Yes	No	No	No	No	No	No
P9	12	Female	Han	Yes	Yes	No	No	No	No	No	No
P10	1.1	Female	Han	No	Yes	Yes	No	Yes	No	Elevated blood urea and uric acid	Elevated liver enzymes
P11	5	Male	Han	Yes	Yes	No	No	No	No	No	No

**Table 2 jcm-12-01428-t002:** Variants of the *PKHD1* gene identified in this study.

Patient	Nucleotide Change	Amino Acid Change	Hom/Het	Location	Domain	Mutation Type	ACMG	Reported
P1	c.2141-3T>C	-	Het	I21	-	Splicing	VUS (PM2 + PP3 + PM5)	No
P2, P9	c.7942G>A	p.Gly2648Ser	Het	E50	Extracellular	Missense	VUS (PM2)	[23]
P2	c.7351-2A>T	-	Het	I46	-	Splicing	P (PVS1 + PM2 + PM5)	[24]
P3	c.3500T>C	p.Leu1167Pro	Het	E30	IPT/TIG 6; atypical	Missense	VUS (PM2 + PP3)	No
P3	c.11174+5G>A	-	Het	I61	-	Splicing	VUS (PM2 + PP3)	No
P4	c.325G>A	p.Ala109Thr	Het	E5	IPT/TIG 1; atypical	Missense	LP (PS2 + PM2)	No
P5	c.6001G>T	p.Glu2001 *	Het	E37	G8 1	Nonsense	P (PVS1 + PM2 + PM5)	No
P1, P5, P6	c.2507T>C	p.Val836Ala	Het	E24	Extracellular	Missense	VUS (PM2 + PP3 + PM3)	[25]
P6	c.5869G>A	p.Asp1957Asn	Het	E36	G8 1	Missense	LP (PS2 + PM2 + PP3)	No
P7	c.6245C>T	p.Thr2082Ile	Het	E38	Extracellular	Missense	VUS (PM2 + PP3)	No
P8	c.6910C>T	p.Gln2304 *	Het	E43	PbH1 3′	Nonsense	P (PVS1 + PM2 + PM3)	[26]
P8	c.4199C>T	p.Ser1400Leu	Het	E32	IPT/TIG 9	Missense	VUS (PM2 + PP3)	[26]
P9	c.11525G>A	p.Arg3842Gln	Het	E65	Extracellular	Missense	VUS (PM2 + PP3)	No
P10	c.9780G>A	p.Trp3260 *	Het	E58	Extracellular	Nonsense	P (PVS1 + PM2 + PM3)	No
P10	c.8518C>T	p.Arg2840Cys	Het	E54	G8 2	Missense	LP (PS1 + PM2 + PM3 + PP3)	[27]
P11	c.10756_10759delAACT	p.Asn3586Serfs *22	Het	E61	Extracellular	Frameshift	P (PVS1 + PM2 + PP3)	[28]
P11	c.10072G>A	p.Asp3358Asn	Het	E60	Extracellular	Missense	LP (PS2 + PM2 + PM2 + PP3)	No

Hom: homozygote. Het: heterozygote. *: premature termination codon.

## Data Availability

Strains and plasmids are available upon request. The authors confirm that all of the data involved in the conclusion of the article are present in the article, figures and tables. The data presented in this study are available at *Journal of Clinical Medicine* online. Table 2 contains the genotypes for each individual.

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
