# Peer review of "A Potential Therapy Using Antisense Oligonucleotides to Treat Autosomal Recessive Polycystic Kidney Disease"

_jcm, 2023, doi:10.3390/jcm12041428_

Round 1
Reviewer 1 Report
Brief summary: This article is a study investigating the use of Antisense oligonucleotides (ASO) as treatment for autosomal recessive polycystic kidney disease. the aim of the study is to show that ASOs can be used to treat this genetic disease. Nevertheless, It is a small study with a small specimen of patients but well designed, with well documented methods and technique that has in part proved that ASOs are a possibility as treatment in this patients early in life
Article: the article is well written, language and english is proficient. It was easy to read and methods and techniques used were thoroughly explained.
I have identified only typos :
1. Page 3, Line 137 "... we used the the MaxEntScan..." the word " the " is used twice.
2. Table 1. " Renal dis-function " should be "Renal dysfunction"
Final comment:
I have found this article very interesting, with an important contribution to current knowledge and future potential treatment
Author Response
Response to Reviewer 1 Comments
Point 1: Page 3, Line 137 "... we used the the MaxEntScan..." the word " the " is used twice.
Response 1: Thank you for this important comment. We agree on this point. As you suggested, we have revised the mistakes in our new manuscript.
Point 2: Table 1. "Renal dis-function " should be "Renal dysfunction"
Response 2: Thank you for this important comment. We agree on this point. As you suggested, in the revision, we have revised the title from “Renal dis-function” to “Renal dysfunction”.

Reviewer 2 Report
COMMENTS FOR THE AUTHORS
The article is novel and clearly proposes a therapeutic development pathway for ARPKD, where Anti-sense oligonucleotides (ASOs) and functional studies such as the In Vitro Minigene Assay may in the future provide answers to serious genetic diseases.
The formal explanation of the disease at the clinical and molecular level is adequate.
REVIEW ABBREVIATIONS
Line 50: says end-stage kidney disease (ESRD), should say end-stage renal disease (ESRD)
Line 203: it says eGRF, it should say eGFR (Glomerular Filtration Rate)
Line 101: says “Our results provide with a basis for future in vivo study and possible clinical translation 101 of PKHD1 exon-21 extending approaches for treatment of ARPKD” ; In this paragraph, some objectives must be expressed.
IMPROVEMENT AREAS
308: Tables 1 and 2 can be improved by adding explanatory captions for some abbreviations (eg Hom/Het not clear enough)
Author Response
Response to Reviewer 2 Comments
Point 1: REVIEW ABBREVIATIONS: Line 50: says end-stage kidney disease (ESRD), should say end-stage renal disease (ESRD)
Response 1: Thank you for this important comment. We agree on this point. As you suggested, in the revision, we have revised the title from “end-stage kidney disease (ESRD)” to “end-stage renal disease (ESRD)”.
Point 2: REVIEW ABBREVIATIONS: Line 203: it says eGRF, it should say eGFR (Glomerular Filtration Rate)
Response 2: Thank you for this important comment. We are sorry for the typos’ expressions. As you suggested, in the revision, we have revised the title from “eGRF” to “eGFR (Glomerular Filtration Rate)”.
Point 3: REVIEW ABBREVIATIONS: Line 101: says “Our results provide with a basis for future in vivo study and possible clinical translation of PKHD1 exon-21 extending approaches for treatment of ARPKD” ; In this paragraph, some objectives must be expressed.
Response 3: Thank you for this important comment. As you suggested, in the revision, we have modified it.
Point 4: IMPROVEMENT AREAS: 308: Tables 1 and 2 can be improved by adding explanatory captions for some abbreviations (eg Hom/Het not clear enough)
Response 4: Thank you for this important comment. As you suggested, in the revision, we have added explanatory captions for some abbreviations in Tables 1 and 2.

Reviewer 3 Report
In this manuscript the authors perform sequence analysis on clinical samples from ARPKD patients in order to identify novel mutations affecting the PKHD1 gene. Mutants are classified, and mutations likely to affect splicing are further analyzed by a minigene assay in which exons and flanking splice donor/acceptor sites are ligated between synthetic splice donor and acceptor sequences. The resulting vector is transfected into various cell types, and the resulting messages are amplified and sequenced to determine whether various mutations interfered with splicing. The analysis confirms various splice defects across the patient spectrum. The authors further perform successful tests of the feasibility of anti-sense oligonucleotide therapy to correct splicing defects in this assay system.
The study is relevant to the ARPKD field, as there is currently no therapeutic approach to ARPKD save dialysis and/or renal transplantation.
The study is well-defined, and the results are for the most part clean, well-controlled, and interpretable.
Major concerns
The authors seem to suggest that their results show correction of abnormal mRNA expressed from the full-length PKHD1 gene in HEK293 cells, but I am fairly certain that all experiments are performed using exon fragments in the minigene assay. This should be made clear to the readers.
There are significant concerns about the grammar, syntax, and usage of the English language throughout the manuscript. The authors must use an editing service to correct this.
Minor Concerns
I think that the authors forgot to use the half-black symbol for healthy carriers. For families like #1, is it stated elsewhere whether the parents are known to not be carriers (i.e. the proband’s mutations are de novo), or whether the sequence data is simply lacking for these individuals?
S2b is an important figure but I had a hard time making sense of it at first due to the presence of the bracket. Instead of the bracket, I think it would be helpful if the bases in E46, E48, and E49 in the “upper panel” nucleotide sequence each had their own color that matched to the nucleotide sequence in the “down panel”. It would also be clearer if splice connectors were drawn into the upper panel (i.e. showing splicing from E46 to E48 to E49), and if the 5’ sequence of E48 in the upper panel extended to the TGA that becomes the premature termination codon, and find some way to call out this TGA (something like this for E48: “GTGGATTTACTGTGA---TGGTTT”).
E47 is in the upper panel should also show all of the 5’ bases and 3’ bases that can be seen in the panel A sequencing data (like this for E47: “GGAAGT---GACAAG”)
The same applies to Figure 1 - It isn’t obvious what the brackets are supposed to be showing. I recommend using unique colors to show the bases of interest so that it is obvious where they are in the gene sequence, and where they show up in the transcript sequences shown below the gene sequence.
In Fig S3, panel D - the labels across the top do not match with the legend in the text. I believe that some of these lanes are reversed (e.g. 3rd lane from the left looks like mutant and the 4th lane looks like E47-WT).
The role of cycloheximide in the NMD assay is not clear. Does CHX also inhibit NMD? A brief sentence (with reference) stating what CHX does in this assay would be helpful.
The CHX experiment is an uncontrolled negative result. The investigators should re-do one of the mutants +/- CHX along with a control on a construct known to undergo NMD. Alternatively a re-do of one of the experiments accompanied by a Western showing CHX-dependent inhibition of protein synthesis.
Please review that all mutants are appropriately matched throughout the manuscript. On at least one occasion there is a mismatch: c.2141-3T>C is listed as belonging to P2 in section 3.2, but this mutation corresponds to P1 in Table 2.
Author Response
Response to Reviewer 3 Comments
Point 1: Major concerns: The authors seem to suggest that their results show correction of abnormal mRNA expressed from the full-length PKHD1 gene in HEK293 cells, but I am fairly certain that all experiments are performed using exon fragments in the minigene assay. This should be made clear to the readers.
Response 1: Thank you for this important comment. The functional tests of hybrid minigenes in splicing reporter plasmids, such as pSPL3, have become valuable tools to check the splicing profiles induced by a sequence variation. The laborious constructions of minigenes and the low pSPL3 efficiency precluded a more widespread use and the cloning of large minigenes with several exons. In the revision, we have added explanatory captions for article that our experiments are performed using exon fragments in vitro minigene assay.
Point 2: Major concerns: There are significant concerns about the grammar, syntax, and usage of the English language throughout the manuscript. The authors must use an editing service to correct this.
Response 2: Thank you for this important comment. As you suggested, we have used a paid editing service at https://www.mdpi.com/authors/english.
Point 3: Minor Concerns: I think that the authors forgot to use the half-black symbol for healthy carriers. For families like #1, is it stated elsewhere whether the parents are known to not be carriers (i.e. the proband’s mutations are de novo), or whether the sequence data is simply lacking for these individuals?
Response 3: Thank you for this important comment. We have used the half-black symbol for healthy carriers in the revised Figure S1(below).
Figure S1. Pedigrees of affected families and variants of the probands. Pedigrees of 9 families with PKHD1 pathogenic variants. Squares indicate males, circles females, filled symbols indicate that the individual presented cysts, open symbols indicate healthy individuals, and half black symbols indicate healthy carrier, NA indicates that DNA sample was unavailable for segregation confirmation. Probands are denoted by arrows.
Point 4: Minor Concerns: S2b is an important figure but I had a hard time making sense of it at first due to the presence of the bracket. Instead of the bracket, I think it would be helpful if the bases in E46, E48, and E49 in the “upper panel” nucleotide sequence each had their own color that matched to the nucleotide sequence in the “down panel”. It would also be clearer if splice connectors were drawn into the upper panel (i.e. showing splicing from E46 to E48 to E49), and if the 5’ sequence of E48 in the upper panel extended to the TGA that becomes the premature termination codon, and find some way to call out this TGA (something like this for E48: “GTGGATTTACTGTGA---TGGTTT”). E47 is in the upper panel should also show all of the 5’ bases and 3’ bases that can be seen in the panel A sequencing data (like this for E47: “GGAAGT---GACAAG”) The same applies to Figure 1 - It isn’t obvious what the brackets are supposed to be showing. I recommend using unique colors to show the bases of interest so that it is obvious where they are in the gene sequence, and where they show up in the transcript sequences shown below the gene sequence.
Response 4: Thank you for this important comment. As you suggested, in the Figure 1 and S2b, we have substitute the bracket with the bases with their own colors in the gene sequence, in the transcript sequences and in the “down panel” nucleotide sequence that matched to the nucleotide sequence in the “upper panel” and call out the premature termination codon TGA. Besides, we have modified and showed all the 5’ bases and 3’ bases of E47 in the upper panel in the panel A sequencing data.
Figure 2, panel D and panel E
Figure S2
Point 5: Minor Concerns: In Fig S3, panel D - the labels across the top do not match with the legend in the text. I believe that some of these lanes are reversed (e.g. 3rd lane from the left looks like mutant and the 4th lane looks like E47-WT).
Response 5: Thank you for this important comment. We are sorry for the wrong labels of the lanes in Fig S3, panel D. In the revision, we have modified the labels across the top in the text in Fig S3, panel D.
Figure S3, panel D. Gel electrophoresis of the RT-PCR product of minigene transcripts in and HepG2 cell of c.7351-2A>T and c.11174+5G>A variant. Lane 1: marker; Lane 2: DDW, Lane 3: pSPL3 (263bp), Lane 4: E47-WT (399 bp), Lane 5: c.7351-2A>T (399 bp and 263 bp), Lane 6: E61-WT(1281 bp), Lane 7: c.11174+5G>A (1281 bp and 263 bp).
Figure S3, panel D
Point 6: Minor Concerns: The role of cycloheximide in the NMD assay is not clear. Does CHX also inhibit NMD? A brief sentence (with reference) stating what CHX does in this assay would be helpful.
Response 6: Thank you for this important comment. The abnormal pre-mRNAs with PTCs may be rapidly degraded by the NMD pathway; this has made progress in the fight against many diseases. The cycloheximide (CHX) as one of the de novo protein synthesis inhibitors were used to clarify whether the mutant mRNA was destroyed by nonsense-mediated mRNA decay (NMD). As you suggested, we have added the reference “PMID: 31145732” in the revised MS.
Point 7: Minor Concerns: The CHX experiment is an uncontrolled negative result. The investigators should re-do one of the mutants +/- CHX along with a control on a construct known to undergo NMD. Alternatively a re-do of one of the experiments accompanied by a Western showing CHX-dependent inhibition of protein synthesis.
Response 7: Thank you for this important comment. We agree with the reviewer’s important suggestion. One of the experiment involving in CHX-dependent inhibition of protein synthesis should be re-do. The cycloheximide (CHX) as one of the de novo protein synthesis inhibitors were used to clarify whether the transcripts from the mutant minigenes were vulnerable to NMD. However, as an uncontrolled negative result, the control on a construct known to undergo NMD in ARPKD have yet to be found. Qiuming Gong had revealed that hERG nonsense mutations are subject to nonsense-mediated mRNA decay by treatment with the protein synthesis inhibitor cycloheximide resulted in the restoration of mutant mRNA to levels comparable to that of the wild-type minigene (PMID: 17576861). Maybe detecting the band of mutant proteins in HEK293 cells to show CHX-dependent inhibition of protein synthesis by western blot (WB) analysis to complete the experiment. If necessary, please give us some times to improve this part.
Point 8: Minor Concerns: Please review that all mutants are appropriately matched throughout the manuscript. On at least one occasion there is a mismatch: c.2141-3T>C is listed as belonging to P2 in section 3.2, but this mutation corresponds to P1 in Table 2.
Response 8: Thank you for this important comment. We are sorry for the mismatch about the variant c.2141-3T>C. The manuscript has been thoroughly checked and revised the mistakes in our revised manuscript.

Round 2
Reviewer 3 Report
Section 3.4
There is a problem with the syntax in this paragraph. Lines 276-278 state that certain variants ARE subject to NMD, but the data do not support this conclusion, and the authors in fact conclude that the variants are NOT subject to NMD (as stated in “Conclusions” line 417). I think that this is an error introduced in the process of English editing. The authors need to look at section 3.4 carefully to make sure that the text is accurate, specifically this sentence: “The c.7351-2A>T (p.G2451Vfs*4) and c.11174+5G>A (p.G3386Efs*10) variants were subjected to degradation by the NMD pathway (Figure 3A).” (see lines 276-278).
Also, it should be stated that CHX has been shown to inhibit NMD:
Line 282 “…inhibitor cycloheximide (CHX), which has been shown to inhibit NMD [27].”
Minor grammar issues:
-Line 200 originally stated “…which resulted in emergent laparoscopic huge splenectomy…” and now states “…which resulted in an extensive mergency laparoscopic splenectomy…”. I believe that this is supposed to be “…which resulted in an emergency laparoscopic splenectomy…”
-Line 285 I think “-control” should be eliminated from the end of the phrase so the sentence reads “…may escape from the control of NMD (Figure 3B, 3C).”
-Line 337 I think the word “fibrocystic” should be removed so the sentence reads “The severity of protein dysfunction…”
Author Response
Response to Reviewer Comments
Point 1: Section 3.4: There is a problem with the syntax in this paragraph. Lines 276-278 state that certain variants ARE subject to NMD, but the data do not support this conclusion, and the authors in fact conclude that the variants are NOT subject to NMD (as stated in “Conclusions” line 417). I think that this is an error introduced in the process of English editing. The authors need to look at section 3.4 carefully to make sure that the text is accurate, specifically this sentence: “The c.7351-2A>T (p.G2451Vfs*4) and c.11174+5G>A (p.G3386Efs*10) variants were subjected to degradation by the NMD pathway (Figure 3A).” (see lines 276-278).
Response 1: Thank you for this important comment. We are sorry for the wrong expressions. Lines 276-278 state that c.7351-2A>T (p.G2451Vfs*4) and c.11174+5G>A (p.G3386Efs*10) variants were predicted to subject to NMD, so we designed the experiment to verify this prediction. Our data do not support this prediction, and in fact the variants are not subject to NMD. We have revised the mistakes in our new manuscript.
Point 2: Also, it should be stated that CHX has been shown to inhibit NMD:
Line 282 “…inhibitor cycloheximide (CHX), which has been shown to inhibit NMD [27].”
Response 2: Thank you for this important comment. We agree on this point. As you suggested, in the revision, we have revised the title from “we used a de novo protein synthesis inhibitor cycloheximide (CHX)[27]” to “we used a de novo protein synthesis inhibitor cycloheximide (CHX), which has been shown to inhibit NMD[27] ”.
Point 3: Minor grammar issues: -Line 200 originally stated “…which resulted in emergent laparoscopic huge splenectomy…” and now states “…which resulted in an extensive mergency laparoscopic splenectomy…”. I believe that this is supposed to be “…which resulted in an emergency laparoscopic splenectomy…”
Response 3: Thank you for this important comment. We agree on this point. As you suggested, in the revision, we have revised the title from “…which resulted in an extensive mergency laparoscopic splenectomy…” to “…which resulted in an emergency laparoscopic splenectomy…”.
Point 4: Minor grammar issues: -Line 285 I think “-control” should be eliminated from the end of the phrase so the sentence reads “…may escape from the control of NMD (Figure 3B, 3C).”
Response 4: Thank you for this important comment. We agree on this point. As you suggested, in the revision, we have revised the title from “…may escape from the control of NMD-control” to “…may escape from the control of NMD”.
Point 5: Minor grammar issues: -Line 337 I think the word “fibrocystic” should be removed so the sentence reads “The severity of protein dysfunction…”
Response 5: Thank you for this important comment. We agree on this point. As you suggested, in the revision, we have revised the title from “The severity of fibrocystic protein dysfunction... ” to “The severity of protein dysfunction…”.
